# The N-Terminal Region of Yeast Protein Phosphatase Ppz1 Is a Determinant for Its Toxicity

**DOI:** 10.3390/ijms21207733

**Published:** 2020-10-19

**Authors:** Carlos Calafí, María López-Malo, Marcel Albacar, Antonio Casamayor, Joaquín Ariño

**Affiliations:** Institut de Biotecnologia i Biomedicina & Departament de Bioquímica i Biologia Molecular, Universitat Autònoma de Barcelona, Cerdanyola del Vallès, 08193 Barcelona, Spain; Carlos.Calafi@uab.cat (C.C.); marialopezmalo@gmail.com (M.L.-M.); Marcel.Albacar@uab.cat (M.A.); Antonio.Casamayor@uab.cat (A.C.)

**Keywords:** protein phosphatases, cell growth arrest, hybrid proteins, *Saccharomyces cerevisiae*

## Abstract

The Ppz enzymes are Ser/Thr protein phosphatases present only in fungi that are characterized by a highly conserved C-terminal catalytic region, related to PP1c phosphatases, and a more divergent N-terminal extension. In *Saccharomyces cerevisiae,* Ppz phosphatases are encoded by two paralog genes, *PPZ1* and *PPZ2*. Ppz1 is the most toxic protein when overexpressed in budding yeast, halting cell proliferation, and this effect requires its phosphatase activity. We show here that, in spite of their conserved catalytic domain, Ppz2 was not toxic when tested under the same conditions as Ppz1, albeit Ppz2 levels were somewhat lower. Remarkably, a hybrid protein composed of the N-terminal extension of Ppz1 and the catalytic domain of Ppz2 was as toxic as Ppz1, even if its expression level was comparable to that of Ppz2. Similar amounts of yeast PP1c (Glc7) produced an intermediate effect on growth. Mutation of the Ppz1 myristoylable Gly2 to Ala avoided the localization of the phosphatase at the cell periphery but only slightly attenuated its toxicity. Therefore, the N-terminal extension of Ppz1 plays a key role in defining Ppz1 toxicity. This region is predicted to be intrinsically disordered and contains several putative folding-upon-binding regions which are absent in Ppz2 and might be relevant for toxicity.

## 1. Introduction

The *Saccharomyces cerevisiae* Ppz1 protein phosphatase is a 692-residue protein made of a C-terminal catalytic domain, about 60% identical to Glc7, the catalytic subunit of genuine yeast PP1 (PP1c), and a long N-terminal segment (~350 residues) unrelated to other proteins [1]. The genome of *S. cerevisiae* contains a paralog, *PPZ2*, with a similar structure. The catalytic domains of Ppz1 and Ppz2 are highly conserved, whereas the N-terminal extension is more divergent [2,3]. It was shown that Ppz1 is myristoylated in vivo at its N-terminal Gly [4]. 

While PP1c enzymes are widely distributed among eukaryotes, Ppz phosphatases are found only in fungi [5]. Ppz phosphatases play an important role in the maintenance of monovalent cation homeostasis, although the specific mechanisms can be somewhat different between species [6,7,8,9,10]. In *S. cerevisiae*, this is done by downregulating the influx of potassium through the high-affinity Trk transporters and by repressing the expression of the Na^+^/K^+^-ATPase encoded by the *ENA1* gene [11,12,13,14]. As a result, cells lacking Ppz1 are hypertolerant to Na^+^ and Li^+^ cations and sensitive to conditions or drugs affecting cell wall integrity, such as high temperature or treatment with calcofluor white or caffeine [2,3,12,15,16]. The latter effect has been explained by the increased turgor pressure caused by augmented K^+^ influx in *ppz1* cells [16]. Conversely, a moderate increase in the expression of Ppz1 or the overexpression of the *PPZ2* gene rescues the lytic defect of a *slt2* strain, which is deficient in the MAP kinase that controls the cell wall remodeling pathway [3]. Mutation of *PPZ2* results in little phenotypic changes, although it potentiates the effects caused by the mutation of *PPZ1* [2,12], suggesting that Ppz2 plays a less relevant cellular role. Ppz1 has been also involved in the regulation of endocytic trafficking and ubiquitin turnover [17] and in the dephosphorylation of the Art1 arrestin, a modification that would be required for interaction of Art1 with Mup1 in response to the presence of methionine in the medium [18]. 

Ppz phosphatases are regulated by two inhibitory subunits, Hal3 and Vhs3, which bind to the catalytic domain of the phosphatase [15,19,20,21,22]. The function of Hal3 as regulator of Ppz phosphatases is more relevant in vivo than that of Vhs3. Remarkably, in *S. cerevisiae* and other related fungi, both are moonlighting proteins that are involved in the biosynthesis of coenzyme A (CoA) through the formation of an atypical heterotrimeric phosphopantothenoylcysteine decarboxylase (PPCDC) enzyme [23].

Excessive Ppz1 activity is detrimental to the cell, causing reduced cell proliferation or even a total blockage in the cell cycle at the G_1_-S transition, which is accompanied by delayed expression of the G_1_ phase cyclins Cln2 and Clb5 [4,15,24,25,26]. A more recent genome-wide search using the genetic tug-of-war (gTOW) approach to identify dosage-sensitive genes demonstrated that *PPZ1* is the gene for which the cell shows the lowest dosage tolerance limit [27], indicating that Ppz1 is likely the most toxic protein when overexpressed in budding yeast. Very recent studies have shown that this toxicity results from the interference with diverse cellular functions, including protein translation, metabolic cell signaling and oxidative stress generation [26,28]. The investigation of the impact of Ppz1 in the cell biology might be relevant because Ppz1 has been recognized as a virulence factor in important human pathogenic fungi, such as *Candida albicans* [7] and *Aspergillus fumigatus* [29], albeit this is not a common trait for all fungal pathogens [30,31].

As mentioned above, most work has been devoted to the elucidation of Ppz1 function and regulation, while Ppz2 has received less attention. We assumed that, because Ppz1 toxicity derives from its phosphatase activity [28], the overexpression of Ppz2 might have similar effects. We show here that this is not the case. In contrast, the overexpression of a chimeric protein bearing the catalytic domain of Ppz2 and the N-terminal extension of Ppz1 was as toxic as that of Ppz1, even if its expression level was similar to that of Ppz2. These results indicate that the N-terminal extension of Ppz1 might have a role in the toxicity of the phosphatase that is not shared by the equivalent N-terminal half of Ppz2.

## 2. Results

### 2.1. The N-Terminal Region of Ppz Phosphatases Is Relevant for Toxicity

Ppz phosphatases belong to the type 1 family of PPP phosphatases, represented in *S. cerevisiae* by the single and essential protein Glc7 (Figure 1). Ppz1 and Ppz2 are very closely related in their C-terminal catalytic domain (85.9% identity, 89.9% similarity), whereas they are far more divergent in the N-terminus (32.2% identity, 45.9% similarity). The catalytic domain of Ppz2 is slightly closer to Glc7 (60.4%/73.6%) than that of Ppz1 (57.1%/69.9%). The evidence that the toxic effect of Ppz1 overexpression, derived from its phosphatase activity and the very high similarity between the Ppz1 and Ppz2 catalytic domains, prompted us to investigate whether the overexpression of Ppz2 was also able to affect cell growth.

To this end, we used an episomal plasmid (YEp352), carrying *PPZ2* expressed from its own promoter, and also cloned the *PPZ2* ORF in the episomal pCM190 vector, to allow regulated expression from the *tetO* promoter. In addition, we created a pCM190-derived version of both Ppz1 and Ppz2 phosphatases carrying a HA epitope tag at their C-terminus. As previously reported [28] and shown in Figure 2A, the expression of *PPZ1* from the pCM190 vector fully abolished growth. In contrast, the expression of *PPZ2* from plasmid YEp352 or from the pCM190 vector did not affect growth in a significative way. The expression of the C-terminally HA-tagged versions of the phosphatases reproduced the behavior observed with the untagged proteins. This pattern was confirmed when the tagged versions of the phosphatases were grown in liquid culture (Figure 2B), with Ppz2-expressing cells displaying a growth profile quite close to those carrying an empty plasmid. As mentioned above, the overexpression of Ppz1 results in delayed G_1_-S transition and accumulation of unbudded cells. Figure 2C shows that the percentage of unbudded cells in cultures overexpressing Ppz2 is virtually the same as that of cells bearing the empty plasmid. In contrast, when treated in the same conditions, cells carrying Ppz1 showed a noticeable increase in the number of unbudded cells. Monitoring of the relative levels of tagged Ppz1 and Ppz2 by immunoblot revealed that the expression level of Ppz2 was markedly lower than that of Ppz1 (Figure 2D). We wondered whether the expression levels of Ppz2 could be insufficient to trigger a toxic effect, or even to perform its normal functions. To test the latter possibility, we transformed *ppz1 ppz2* cells, which are sensitive to the temperature of 37 ℃ [3], with the diverse constructs bearing Ppz2. As can be observed in Figure 3, all of them, irrespective of the presence of the tag, were able to restore normal tolerance to high temperature, indicating that the produced protein was indeed functional.

We then created two hybrid Ppz versions, both C-terminally HA-tagged. In the first one (Ppz1:2), the N-terminal region of Ppz1 was followed by the catalytic domain of Ppz2. In the second (Ppz2:1), the N-terminal region of Ppz2 was tailed by the catalytic domain of Ppz1 (see Figure 1 and Appendix A). To our surprise, while the expression of the Ppz2:1 construct behaved as Ppz2, that is, nearly without effect on growth, the expression of the Ppz1:2 version blocked proliferation in a similar way as Ppz1 did (Figure 2A,B). Similarly, the accumulation of unbudded cells expressing the Ppz1:2 chimera was indistinguishable from those expressing Ppz1 whereas, as mentioned above, cultures of cells bearing Ppz2 did not show alteration in the percentage of unbudded cells (Figure 2C). In addition, the recently reported beneficious effect on growth of the *gcn2* [28] and *hog1* [26] mutations in cells overexpressing native Ppz1 was mimicked in cells expressing the Ppz1:2 hybrid phosphatase (Appendix A). When the expression of these hybrid versions was investigated, we observed that none of them matched that of Ppz1, but while that of Ppz2:1 was very low, the expression of Ppz1:2 was similar to that observed for Ppz2. Therefore, this was indicative that the expression of the Ppz1:2 construct was toxic even at levels clearly lower than that of native Ppz1 and matching those of Ppz2 expression. It must be noted, however, that in spite of the lack of effect on cell growth, the Ppz2:1 construct generated a functional protein since, as documented in Figure 3, it was able to restore normal thermotolerance in Ppz-deficient cells.

To further test the ability of the tagged versions of Ppz2 and Ppz2:1 to supply normal Ppz2 function, we verified the ability of these proteins to normalize the tolerance to lithium and caffeine of a double *ppz1 ppz2* mutant. This strain is highly tolerant to Li^+^ ions and hypersensitive to caffeine, and such phenotypes are already clearly detected in the single *ppz1* mutant (Figure 4). As can be observed in Figure 4, the tagged versions of Ppz2 and Ppz2:1 were nearly as effective as native *PPZ2* in normalizing tolerance to caffeine, whereas they only displayed a very mild effect when cells were challenged with lithium.

### 2.2. Mutation of Myristoylable Gly2 in Ppz1 only Slightly Attenuates Toxicity

Ppz1 can be myristoylated in vivo and in vitro at its N-terminal Gly2. Due to the relevance of this covalent modification in the subcellular localization of proteins, we sought to investigate if this feature might be relevant for the toxicity associated to the overexpression of the phosphatase. As shown in Figure 5A, the expression of a version of Ppz1 in which Gly2 has been changed to Ala (G2A) allowed slight but visible growth in dot tests, whereas the growth of cells expressing Ppz1 was undetectable. A slight improvement in growth in the case of the G2A version can be also observed in liquid culture experiments (Figure 5B). Evaluation by immunoblot of the expression of this variant showed that it is produced at levels comparable to those of native Ppz1 (Figure 5C), suggesting that the somewhat lesser toxic effect of the G2A variant could not be explained by lower expression levels. It was known that Ppz1 can be found at the plasma membrane [11,18], so we tested if this was the case when a C-terminally GFP-tagged version was overexpressed from an episomal plasmid. Phenotypic testing of cells expressing the Ppz1-GFP-tagged protein showed that this variant was indistinguishable from the native phosphatase (Appendix A). As it can be observed in Figure 5D, a significant fluorescent signal is detected at the periphery of the cells when native Ppz1 is present. In contrast, when the G2A version is expressed from the same plasmid, Ppz1 disappears from the cell’s periphery and fluorescence is widely distributed inside the cell, indicating that the lack of the myristoylable Gly affects the subcellular localization of the phosphatase.

### 2.3. Comparison of the Effect of Glc7 and Ppz2 Expression

It has been described that the overexpression of Glc7 at high levels is toxic to the cells. We wanted to compare the relative effects of the expression of Glc7, Ppz1, and Ppz2 derived from the same pCM190 expression vector. Because the available Glc7 version was tagged at the N-terminus, we decided to create a novel N-terminally-tagged version of Ppz2 for better comparison. As is shown in Figure 6A, the expression of N-terminally-tagged Ppz2 did not affect growth when cells were grown on glucose or galactose. In contrast, the expression of Glc7 reduced growth on glucose and fully blocked proliferation on galactose. The growth reduction imposed by the overexpression of Glc7 in cells growing on glucose was also noticeable in liquid cultures (Figure 6B). When the levels of these proteins were evaluated by immunoblot, it was found that HA-Glc7 was expressed at somewhat lower levels than C-terminally-tagged Ppz1 (Figure 6C). In contrast, the expression levels of HA-Glc7 and HA-Ppz2 were rather alike. The phenotypic characterization of cells expressing the HA-Ppz2 version showed (Appendix A) that it can effectively rescue growth at 37 ℃ to wild type levels, not only of the *ppz1 ppz2* strain (note that these cells grow better than the single *ppz1* mutant carrying an empty plasmid), but also that of a *slt2* strain, which is thermosensitive because a defect in cell wall construction. Therefore, the HA-Ppz2 protein seems to fulfil successfully the role of native Ppz2.

### 2.4. Comparative Analysis of Ppz1 and Ppz2 N-Terminal Regions

In view of the dramatic difference in the effect of the overexpression of Ppz1 and the Ppz1:2 hybrid in comparison with Ppz2, we focused our attention on the structural differences of the N-terminal extensions of both phosphatases. The exchanged fragments were 349 (N-Ppz1) and 383 (N-Ppz2) residues long (Figure 1 and Appendix A). Analysis of the amino acid composition of these regions (Appendix A) reveals, in both cases, a high number of Ser (26.1% Ser in N-Ppz1, 22.7% in N-Ppz2), which represents a 2- to 3-fold excess over the average yeast proteome composition, a prevalence of basic residues, mostly Arg (12% Lys + Arg in N-Ppz1, 14.1% in N-Ppz2), and a relatively scarcity of acidic residues (mostly Glu), which results in a relatively high pI (9.44 for N-Ppz1, 10.28 for N-Ppz2). Therefore, within this context, both regions share rather common features. When the sequence of both proteins was analyzed with the IUPred2A predictor in search of intrinsically disordered regions, it became evident that while their C-terminal halves were strongly structured, the N-terminal sections showed the characteristic traits of intrinsically disordered regions (IDRs, Figure 7A), with virtually the entire N-terminal segment of both proteins above the 0.5 threshold score. In contrast, when the existence of regions able to adopt a structure upon interaction with other proteins was examined with the Anchor2 algorithm (Figure 7B), the profile was more divergent, with six regions in Ppz1 within residues 50 to 340 (annotated from I to VI), with scores reaching the 0.5 threshold, most of them not predicted in Ppz2.

## 3. Discussion

We show in this work that, in contrast to Ppz1, the overexpression of Ppz2 from a multicopy plasmid, either driven from its own promoter or a *tetO*_7_ promoter, only produces a very mild effect on yeast cell growth. This difference fits with the results obtained by Makanae and coworkers [27] who, on the basis of a gTOW screen, assigned a copy number limit about 30-fold higher to *PPZ2* than to *PPZ1*. The high toxicity of overdosed *PPZ1* when compared to *PPZ2* likely explains why the former was not identified in the early high-copy suppressor screen of the lytic phenotype of the *slt2*/*mpk1* mutant in which *PPZ2* emerged [3]. 

Our results show that, under the same conditions for expression, the levels of Ppz2 are lower than that of Ppz1, a fact that could be observed for both N- and C-terminally-tagged versions of Ppz2 (compare Figure 2D and Figure 6C). This raised the possibility that the amount of Ppz2 protein generated could be insufficient not only to cause a toxic effect but also to fulfill its native functions. However, we show that, in all cases, the amount of protein generated was sufficient to restore normal tolerance to high temperature and to caffeine in a double *ppz1 ppz2* mutant. Remarkably, the untagged *PPZ2* version was also able to restore near wild type tolerance to LiCl, whereas the C-terminally (Figure 4) or the N-terminally-tagged (not shown) versions expressed from the *tetO* promoter had little effect. We have not a definitive explanation for this observation, but it is worth noting that this dissociation of the effects related to caffeine and salt tolerance (LiCl or NaCl) is not totally surprising, since it has been found for diverse versions of Ppz1 carrying mutations at the N-terminal half (see below). On the other hand, it is unlikely that the effective suppression of lithium tolerance could be due to a higher level of *PPZ2* expression from its native promoter, compared to the *tetO*_7_-driven promoter. It is known that the expression of Ppz1 from its native promoter is higher than that of Ppz2 [32], and we observe that the impact in cell growth is more potent when *PPZ1* is expressed from the pCM190 vector than from its own promoter. Therefore, it is tempting to speculate that the presence of the HA tag might be affecting the capacity of Ppz2 to regulate tolerance to Li^+^ cations by an unknown mechanism. In fact, the slight attenuation in toxicity observed in the C-tagged version of Ppz1 (compare profiles of Figure 2B and Figure 5B) suggests that the presence of this tag might not be totally neutral.

The evidence that the lack of toxicity of Ppz2 is not due to lower cellular amounts of the protein can be derived from the observation that the hybrid Ppz1:2 protein is as toxic as native Ppz1 even if its expression level is roughly equal to that of Ppz2. In addition, we show that the expression of Glc7 has a clear negative impact on cell growth compared with that of Ppz2, even if both proteins are expressed at approximately the same levels (Figure 6B,C). We also observe that the overexpression of the Ppz2:1 hybrid, bearing the Ppz2 N-terminal half, has little or no effect. Although it must be recognized that the level of expression of the Ppz2:1 chimera is lower than that of Ppz1:2 (Figure 2D), we feel that the difference in expression hardly justifies such a drastic alteration in phenotype, implying that the Ppz2:1 hybrid protein would not be toxic even if it would be expressed at higher levels. On the other hand, the molecular bases for the toxicity Ppz1:2 protein are likely the same as that of native Ppz1, as deduced from their quantitatively identical impact on cell budding (Figure 2C) and the identical growth improvement caused by the *gcn2* and *hog1* mutations (Appendix A).

Therefore, these experiments implicate the N-terminal extension of Ppz1 as a key component in the effect in cell growth when the enzyme is expressed at high levels. Several lines of evidence suggest that such an effect is not due to the mere increase in the amount of this N-terminal polypeptide sequence but would derive from the modulation of the Ppz1 phosphatase function or/and activity. Indeed, (i) the phosphatase catalytic activity of Ppz1 is necessary for toxicity [28], whereas the overexpression of the *S. cerevisiae* N-terminal half alone (from a high-copy plasmid and driven by the strong *GAL* promoter) has no effect on cell growth [4]; and (ii) there is plenty of evidence that the modification of the N-terminal half of Ppz1 has important and specific effects in Ppz1 function. Thus, a large deletion affecting from amino acid 17 to 193 in Ppz1 resulted in the normal complementation of the LiCl tolerance phenotype but caused significant loss of the ability to restore normal caffeine sensitivity, whereas the removal of residues from 241 to 318 had the opposite effect. It is worth noting that the former deletion eliminated the region containing the predicted disordered binding regions I, II, and III, shown in Figure 7B, and that the latter eliminated the V_a,b_ region. The functional relevance of the N-terminal region was also observed in the salt tolerant yeast *Debaryomyces hansenii*, where two short deletions at the N-terminus of DhPpz1 (spanning residues 27−36 and 106−120) abolished the ability to complement LiCl tolerance without affecting the recovery of normal caffeine tolerance [10]. In contrast, no difference with the native CaPpz1 was observed for these phenotypes when three different deletions (25−43, 67−108, and 120−142) of the *C. albicans* phosphatase were tested, even if the in vitro activity of the 25−43Δ version was found to be significantly higher than that of the native enzyme [33]. 

An important modification of the N-terminus of Ppz1 is the mutation of its Gly2, leading to the inability of the protein to be myristoylated [4]. The mutation of Gly2 to Ala affects certain functions of Ppz1 but not all. For instance, it was reported that such a mutation barely affects the ability to complement the caffeine-sensitive phenotype of the *ppz1* mutant in *S. cerevisiae*, but does not allow restoring normal tolerance to Li^+^ [4], and a parallel situation was observed for the *C. albicans* protein [33]. Similarly, Lee and coworkers recently reported [18] that, while the Ppz1-G2A variant was unable to complement the endocytic-trafficking defects found in *ppz1* cells, or the canavanine-hypersensitive phenotype of a *ppz1 ppz2* double mutant strain, it fully complemented the ubiquitin-deficient phenotype previously described for cells lacking Ppz1 [17]. On the other hand, it is generally accepted that myristoylation allows certain proteins to localize to membranes. We show here that, in agreement with previous reports [11,18], native Ppz1 appears mostly located at the cell periphery, likely at the plasma membrane, and shows a punctuated pattern, with less evident cytosolic distribution. We also confirm here that the G2A version of Ppz1 no longer appears at the cell periphery, but shows a diffuse distribution almost within the entire cell [18]. Therefore, since the inability to undergo myristoylation drastically affects the subcellular localization of Ppz1, but the modified enzyme retains some functions, it must be concluded that only a subset of the cellular roles of Ppz1 require localization at the plasma membrane. We also show here that the G2A mutant retains most of the characteristic toxicity of native Ppz1 when overexpressed. This suggests that the major targets for Ppz1 toxicity do not localize at the plasma membrane, a hypothesis that fits well with our recent proposal that Ppz1 overexpression negatively impinges the protein translation process [28] and with the identification of two major cytosolic protein kinases, Snf1 and Hog1, as likely targets for the development of Ppz1 toxicity [26].

Similarly to Ppz1, the overexpression of Glc7 has been revealed to be deleterious for the cell [34,35]. However, a number of evidences suggest that the mechanisms for toxicity are not the same [see [28] for a discussion]. We show here that the expression of Glc7 from a pCM190 vector negatively affects cell growth, although not as strongly as that of Ppz1. However, in this case, it cannot be ruled out that the lower level of Glc7 expression might explain, at least in part, the lesser toxicity observed. In any case, it is interesting to note (Figure 6A) that the growth defect in cells overexpressing Glc7 is exacerbated when the carbon source is galactose (no growth was observed even after 6 days of incubation, not shown). This could be attributed to the reported role of Glc7 in dephosphorylating and inactivating Snf1 and Mig1 (a transcriptional repressor downstream Snf1), thus interfering with the mechanisms that permit adaptation to glucose scarcity and alternative carbon sources [36]. 

Our finding that the N-terminal half of Ppz1 is an important determinant for toxicity adds a further layer of complexity in the mechanism of regulation of these phosphatases and indicates that relevant differences exists between Ppz1 and Ppz2 in these regions. As shown in Figure 7, both the Ppz1 and Ppz2 N-terminal halves markedly show the profile of IDRs. It is commonly accepted that IDRs are enriched in proteins involved in signaling and regulatory functions (see ref. [37] and references therein) and that they are able to mediate functional protein-protein interactions. This is often done by molecular recognition features (MoRFs) or short linear motifs, which usually undergo disorder-to-order transitions upon binding to their partners. Analysis of Ppz1 and Ppz2 N-terminal extensions indicates that Ppz1 contains several short regions for which a significant tendency to drive protein-protein interactions is predicted that are not present in Ppz2. This raises the possibility that the toxic effect of Ppz1 might be the result of a specific association of the phosphatase with key targets mediated by its N-terminal half. Such a hypothesis could be tested by identifying specific Ppz1 and Ppz2-binding proteins in affinity-purification or co-immunoprecipitation experiments. We have subjected the N-terminal region of Ppz proteins from diverse fungi to Anchor2 analysis and found that, whereas they are, in general, largely made by IDRs, not all of them contain predicted folding-upon-binding regions (Appendix A). In particular, the somewhat shorter N-terminal extensions of both *C. neoformans* isoforms, as well as the single Ppz1 from *A. nidulans,* lack this characteristic. This is suggestive, since we have recently shown that none of these proteins causes toxicity when overexpressed in *S. cerevisiae,* [30,31]. In the case of *C. neoformans* we postulated, on the basis of the existing evidence, a lower specific activity and/or the absence of key cellular targets as reasons for the absence of toxicity. However, our current results provide an alternative explanation. Further investigation will be needed to precisely dissect the key residues at the N-terminal half of Ppz1 and the hypothetic interacting proteins responsible for this behavior. 

## 4. Materials and Methods

### 4.1. Yeast Strains

Yeast cells were incubated at 28 °C (unless otherwise stated) in YP medium (1% yeast extract, 2% peptone) or in synthetic medium (SC) lacking uracil [38], supplemented with glucose or galactose at 2% as carbon source. Plates contained 2% agar. Yeast cells transformed with pCM190-derived constructs were always plated in medium containing doxycycline (100 µg/mL, note that this concentration is higher than usually employed with this promoter). With the exception of strain DBY746 (*MAT*α, *his3-*Δ*1*, *leu2−3*,*112*, *ura3−52*, *trp1−289*), used for fluorescence microscopy, all yeast strains used in this work were derived from BY4741 (*MAT* a *his3*∆1 *leu2*∆ *met15*∆ *ura3*∆) [39]. The *ppz1*Δ and *slt2*Δ strains are kanMx4 deletant from the EUROFAN collection. The *gcn2* strain is a kanMX4 deletant and the *hog1* strain is a natMX4 deletant, described in [28] and [26], respectively. Strain AGS19 is a *ppz1::KANMx4 ppz2::HIS3* double mutant and was described earlier [40]. Disruption of *PPZ1* with the *LEU2* gene was made is the same way that the previously reported AGS9 strain [41] but using the DBY746 genetic background, giving rise to strain RSC52.

### 4.2. Growth Conditions

For growth on plates, cells were cultured overnight in synthetic medium lacking uracil with glucose 2% as the carbon source (pCM190 vectors in the presence of 100 μg/mL doxycycline), and collected by centrifugation. Cells were resuspended in the same medium without the antibiotic and growth resumed for 5 h. Then, cultures were diluted to OD_600_ = 0.05 and spotted (plus 1/5 dilutions) in plates with or without doxycycline. 

For growth determination in liquid, cultures cells were grown overnight in the presence of doxycycline as above and, after centrifugation, were washed twice in medium lacking doxycycline and resuspended to an OD_600_ = 0.2. Growth was resumed for 8 h and then samples were diluted to OD_600_ = 0.004 (except indicated otherwise) and distributed by triplicate in honeycomb plates (Thermo Fisher Sci., Waltham, MA, USA). Growth was monitored in a BioScreen C apparatus (Thermo Fisher Sci., Waltham, MA, USA) with OD_600_ readings each 30 min.

For the determination of the budding index, BY4741 cells transformed with the relevant plasmids were processed as for growth determination in liquid culture. After 8 h growth in the absence of doxycycline, the culture was diluted to OD_600_ = 0.02 with fresh medium and growth in the absence of the antibiotic resumed for 16 additional h. Then, cells were fixed with 2% formaldehyde and pictures were taken with a Nikon Eclipse TE2000-E microscope (×1000). Samples were anonymized, cells were independently classified into budded or unbudded by at least two persons, individual counts were averaged, and the percentage of unbudded vs. total number of cells was calculated. 

### 4.3. Plasmid Construction

*Escherichia coli* DH5α cells were used as plasmid DNA host and were grown at 37 °C in LB medium supplemented with 50 μg/mL ampicillin, when carrying plasmids. Transformations of *S. cerevisiae* and *E. coli* and standard recombinant DNA techniques were performed as described [42]. 

For pCM190-PPZ2 construction, the *PPZ2* ORF was amplified from BY4741 genomic DNA with oligonucleotides OMLM7 y OMLM8 and the product (2.1 Kbp) was cloned at the KpnI and PstI sites of vector pCM190 (2-micron, *tetO*_7_-driven promoter) [43]. The pCM190-HA-PPZ2 plasmid was made in the same way but using primer HA_PPZ2_Fw_KpnI instead of OMLM7. For plasmid pCM190-HA-GLC7, the GLC7 ORF was amplified with primers HA_GLC7_Fw_BamHI y OMLM11 and the 1.5 Kbp product cloned into the BamHI and PstI sites of pCM190.

Vector pCM190-PPZ1, expressing untagged Ppz1, was constructed, as described in [28], using oligonucleotides OMLM3 and OMLM4. The untagged pCM190-PPZ1^G2A^ version was made in the same way but using oligonucleotides OMLM9 and OMLM4 for amplification. Plasmid pCM190-PPZ1-HA was made by PCR amplification of the *PPZ1* ORF from BY4741 genomic DNA with oligonucleotides OMLM3 and OCCP1. The amplification product was cloned into the BamHI and PstI sites of the vector. pCM190-PPZ2-HA was constructed similarly by amplification of the *PPZ2* ORF with oligonucleotides OMLM7 and OCCP2 followed by cloning of the amplification fragment into the KpnI and PstI sites of pCM190. Generation of pCM190-PPZ1:2-HA was as follows. The C-terminal region of *PPZ2* (from nucleotide 1153 to the HA epitope tag) was amplified using pCM190-PPZ2-HA and oligonucleotides PPZ2-Cter-BspEI-Fw and OCCP2. Then, the 1.0 Kbp product was cloned into the BspEI and PstI sites of pCM190-PPZ1-HA. Finally, plasmid pCM190-PPZ2:1-HA was made by amplification of the N-terminal region of *PPZ2* (from the initiating ATG until nt 1152) using plasmid pCM190-PPZ2-HA as template and primers OMLM7 and PPZ2_Nter_BspEI_Rv. The resulting PCR product (1.2 Kbp) was cloned into the KpnI and BspEI sites of pCM190-PPZ1-HA.

Vectors expressing C-terminally GFP-tagged versions of Ppz1 and PPZ1^G2A^ were constructed starting from strain (DVS001), containing a *PPZ1-GFP* genomic integration, which was previously made as follows. Wild type strain DBY746 was transformed with a cassette amplified from the vector pYM28 with oligonucleotides S3_Ppz1_Ct_Fluo_Fw and S2_Ppz1_Ct_Fluo_Rev, as described in [44]. For generation of YEp195-PPZ1-GFP, the *PPZ1-GFP* ORF was amplified from the genome of strain DVS001 using oligonucleotides PPZ1_Fw_−470_KpnI and EGFP_mCherry_Rv_HindIII. The amplified fragment was cloned into a KpnI and HindIII-digested YEplac195 (2-micron, *URA3*) plasmid [45]. For YEp195-PPZ1^G2A^-GFP, Gly2 was mutated to Ala by the Quickchange^TM^ strategy (Agilent) using YEp195-PPZ1-GFP as template and oligonucleotides Ppz1 G2A_Fw and Ppz1 G2A_Rv. The Expand High-fidelity DNA polymerase (Roche, Sant Cugat, Spain) was usually used for PCR amplification and the Q5^®^ High-Fidelity DNA Polymerase (New England Biolabs, Ipswich, MA, USA) for Quickchange^TM^ mutagenesis. All products were sequenced to detect unwanted mutations. 

Plasmids YEp195-PPZ1 and YEp352-PPZ2 are described in [46] and [3], respectively, and the sequence of oligonucleotides mentioned in this work can be found in Appendix A. 

### 4.4. Preparation of Extracts and Immunoblotting

Cells containing pCM190-derived plasmids were grown o/n in the presence of doxycycline (100 µg/mL). Usually, cultures were centrifuged and resuspended in the same medium lacking doxycycline to an OD_600_ of 0.2, and growth was resumed for 7 h. In the case of cells carrying untagged Ppz1 or Ppz1^G2A^, cells were resuspended at OD_600_ of 0.04 and growth resumed for 16 h. In all cases, extracts were prepared as follows. Cell pellets were resuspended in 125 µL of Lysis Buffer (50 mM Tris-HCl pH 7.5, 150 mM NaCl, 0.1% Triton X−100, 10% glycerol) supplemented with EDTA-free Protease Inhibitor Cocktail (Roche, Sant Cugat, Spain) and 2 mM dithiothreitol. Cells were disrupted by vigorous shaking after adding 125 µL of Zirconia beads (0.5 mm) in a FastPrep cell breaker at setting 5.5 for 45 s (3 cycles). Samples were centrifuged at 500× *g* for 10 min at 4 °C and the total protein of the cleared supernatants quantified by the Bradford method (Sigma-Aldrich, Madrid, Spain). When stated, supernatants were subjected to a second centrifugation step at 25,000× *g* for 30 min at 4 ℃. Proteins were subjected to SDS-PAGE (8% gels) and transferred onto polyvinylidene difluoride (PVDF) membranes (Immobilon-P, Millipore, Mollet del Vallès, Spain). For HA-tagged proteins, membranes were probed with anti-HA antibodies (Roche, Sant Cugat, Spain orBiolegend, San Diego, CA, USA), washed, and incubated with 1:10,000 dilution of secondary anti-mouse IgG-horseradish peroxidase antibodies (GE Healthcare, Chicago, IL, USA). For the detection of untagged Ppz1 or Ppz1^G2A^, membranes were probed with polyclonal anti GST-Ppz1 antibodies (1:250 dilution), followed by a 1:20,000 dilution of secondary anti-rabbit IgG-horseradish peroxidase antibodies (GE Healthcare, Chicago, IL, USA). Immunoreactive proteins were detected with the ECL Prime Western blotting (GE Healthcare, Chicago, IL, USA) detection kit. Membranes were stained with Ponceau Red to monitor proper loading and transfer.

### 4.5. In Vivo Fluorescence Microscopy

To monitor the subcellular localization of native Ppz1 and G2A versions, DBY746 cells were transformed with plasmids YEp195-Ppz1-GFP or YEp195-PPZ1^G2A^-GFP, carrying a C-terminal GFP tag. Single colonies from each transformation were grown overnight at 28 ℃ in SC lacking uracil. Saturated cultures were diluted in the same medium to OD_600_ = 0.2 and growth resumed until OD_600_ = 0.6−0.8 was reached. At this point, the cultures were diluted again to OD_600_ = 0.12 and 300 µL of each culture were deposited into µ-Slide 8 Well chambers (Ibidi^®^). Images were taken using a Nikon Eclipse TE2000-E microscope and the FITC (ex: 480/30 nm, em: 535/45 nm) filter.

### 4.6. Other Methods

The percentage of identity between Glc7, Ppz1, and Ppz2 was calculated with the EMBOSS Needle tool (https://www.ebi.ac.uk/Tools/psa/emboss_needle/, accessed on 14 April 2020), which is based in the Needleman–Wunsch alignment algorithm [47]. Aminoacidic composition and pI of specific protein sequences were calculated at the ExPASy server (https://web.expasy.org/compute_pi/, accessed on 25 May 2020). Prediction of intrinsically disordered regions (IDRs) was done with IUPred2 software, a biophysics-based model (with the ‘short disordered regions’ setting), and putative disordered binding regions were detected using the ANCHOR2 prediction algorithm [48]. The calculation of the amino acid distribution in the yeast proteome was done with an in-house algorithm implemented in Visual Basic using the “orf_trans.fasta.gz” file (2019−10−25) from Saccharomyces Genome Database (http://sgd-archive.yeastgenome.org/sequence/S288C_reference/orf_protein/, accessed on 25 May 2020). This file contains the translations of all systematically named ORFs, except “Dubious” ORFs and pseudogenes.

## Figures and Tables

**Figure 1 ijms-21-07733-f001:**
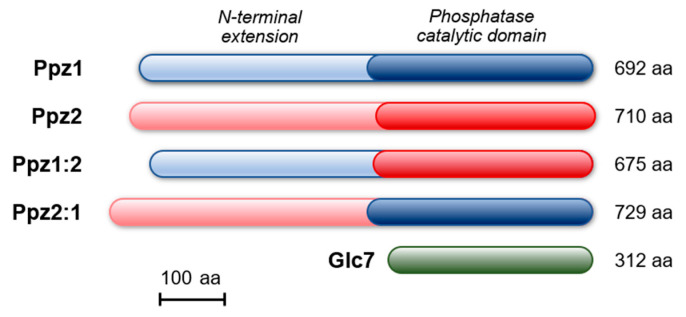
Cartoon depicting the structure and size of Ppz1, Ppz2, the hybrid Ppz proteins and Glc7. Ppz1:2 denotes a construct composed of the N-terminus of Ppz1 and the C-terminus of Ppz2. Ppz2:1 indicates the opposite construct. The size of the native and recombinant proteins is indicated on the right and does not include the 1xHA epitope tag present in some constructs.

**Figure 2 ijms-21-07733-f002:**
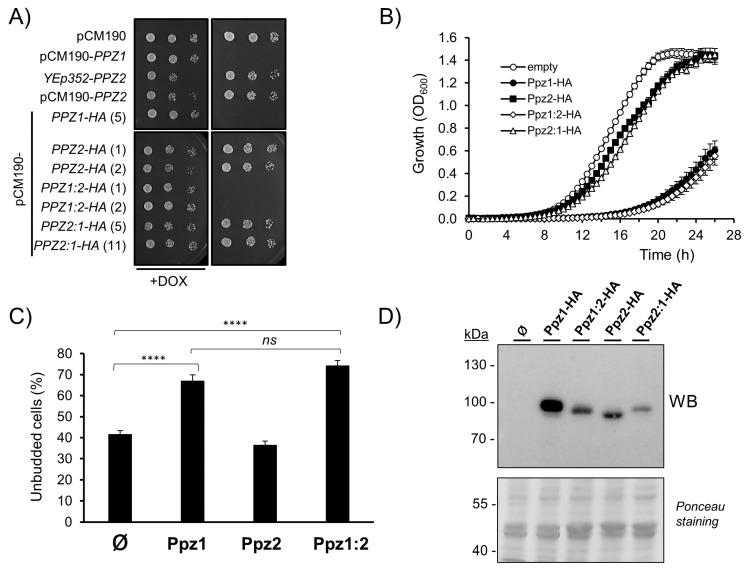
Effect on cell growth of expression of Ppz1, Ppz2, Ppz1:2, and Ppz2:1 proteins. (**A**) Cultures of BY4741 cells transformed with the indicated plasmids were grown as indicated in Materials and Methods and spotted at OD_600_ = 0.05 (plus 1/5 dilutions) on synthetic medium plates lacking uracil and with 2% glucose as carbon source, with (+DOX) or without doxycycline (100 µg/mL). Numbers in parentheses indicate specific clones tested. Pictures were taken after 48 h of growth. (**B**) BY4741 cells transformed with the pCM190-derived plasmids used above were grown in liquid medium as described in Materials and Methods and the OD_600_ determined every 30 min. Data is presented as the mean ± SEM of at least three independent experiments made by triplicate. The clones used were 5 (Ppz1), 1 (Ppz2), 1 (Ppz1:2), and 5 (Ppz2:1). (**C**) Budding index determination. Cells carrying the indicated plasmids (pCM190-derived, HA-tagged versions, Ø denotes the empty vector) were processed as described and the percentage of unbudded vs. total cells determined. Data are mean ± SEM of 4−6 determinations from 3 independent experiments. At least 100 cells/sample and experiment were counted. **** *p* < 0.0001; ns, not significative (*p* > 0.05) as assessed by unpaired *t*-test. (**D**) Cultures of cells carrying the indicated constructs were collected and protein extracts prepared. Cleared supernatants (650 μg of protein) were subjected to centrifugation (25,000× *g* for 30 min at 4 ℃) and the pellets were recovered, resuspended in 1× SDS-PAGE buffer, electrophoresed (8% gels), and transferred to membranes. Staining of the membrane by Ponceau Red for protein detection is shown in the lower panel.

**Figure 3 ijms-21-07733-f003:**
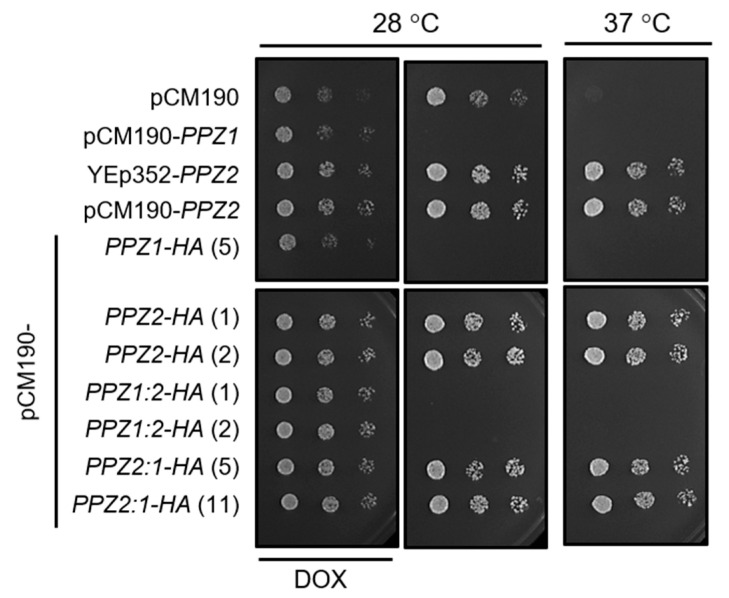
Expression of Ppz2 and Ppz2:1 counteract the heat-sensitive phenotype of Ppz-deficient cells. Strain AGS19 (*ppz1 ppz2*) was transformed with the indicated plasmids. Cultures were spotted, as described for Figure 2A, and plates were incubated for 48 h at the indicated temperatures.

**Figure 4 ijms-21-07733-f004:**
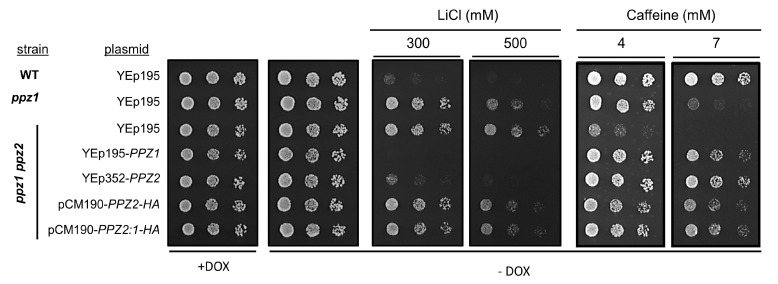
Effect of Ppz2 and Ppz2:1 expression in the lithium-tolerant and caffeine-sensitive phenotype of Ppz-deficient cells. Wild type BY4741 (WT), ppz1Δ and AGS19 strains were transformed with the indicated plasmids and spotted, as in Figure 2A, on plates containing the indicated amounts of LiCl or caffeine. Plates were incubated for 3 days and pictures taken. Clones used for *PPZ2-HA* and *PPZ2:1-HA* were 1 and 5, respectively.

**Figure 5 ijms-21-07733-f005:**
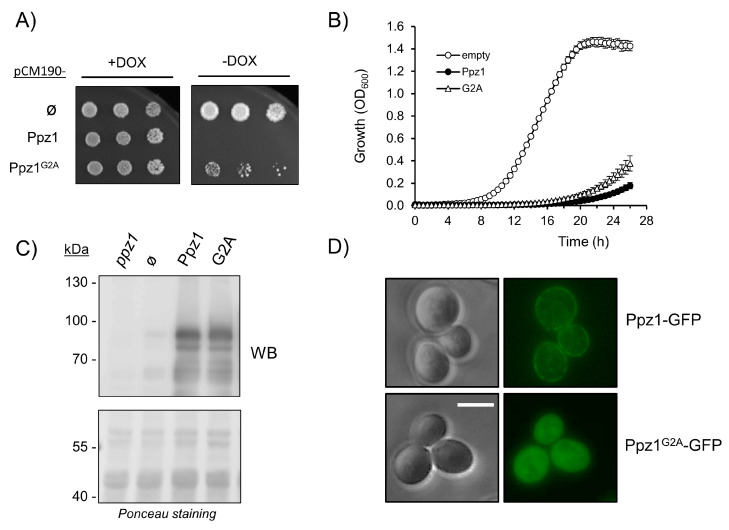
Analysis of the G2A Ppz1 variant. (**A**) Cultures of BY4741 cells transformed with the indicated pCM190-derived plasmids were spotted, as in Figure 2A, and pictures were taken after 6 days. Ø denotes the empty vector (**B**) The same cells were grown in liquid culture, as in Figure 2B. Data correspond to the mean ± SEM of 4−6 determinations. (**C**) Protein extracts of wild type strain BY4741 carrying the indicated plasmids were prepared as described in Materials and Methods and 40 μg of protein were electrophoresed in 8% gels, transferred to membranes, and probed with anti-Ppz1 polyclonal antibodies. The *ppz1* derivative is included as negative reference. Ponceau staining of the membrane is shown in the lower panel. (**D**) Strain DBY746 was transformed with the indicated YEp195-based constructs, and pictures were taken at 1000× amplification, as described in Materials and Methods. Scale bar corresponds to 5 μm.

**Figure 6 ijms-21-07733-f006:**
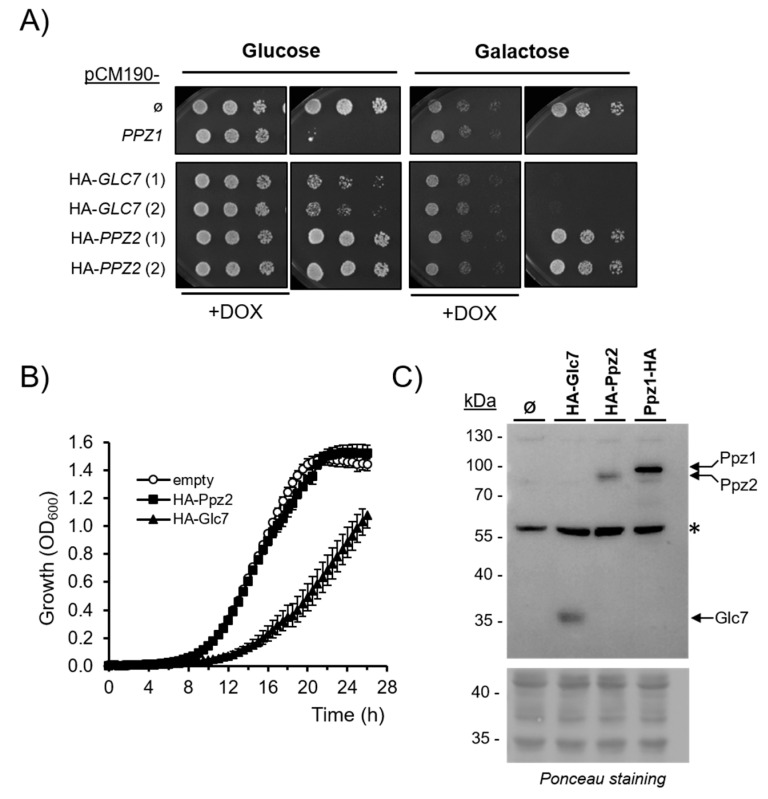
Effects on growth of expression of N-terminally-tagged Glc7 and Ppz2-phosphatases. (**A**) Wild type strain BY4741 was transformed with the indicated plasmids and spotted on plates containing the indicated carbon sources (2%). Ø, denotes the empty pCM190 plasmid. *PPZ1* is an untagged version of the phosphatase. Plates were incubated for 3 days and all clones for a given condition were in the same plate. Numbers denote the clones tested. (**B**) Cells carrying the indicated plasmids were processed as in Figure 2B. Data are mean ± SEM of 4−6 determinations. (**C**) Protein extracts of BY4741 cells transformed with the indicated plasmids (clones 1 for Glc7 and Ppz2) were obtained as described in Materials and Methods. Cleared supernatants (500× *g*) were subjected to SDS-PAGE (40 μg of protein, 10% gels). After transfer to membranes, immunoreactive proteins were identified with anti-HA antibodies. The asterisk indicates a non-specific band recognized by the antibody. Ponceau staining of the membrane is shown in the lower panel.

**Figure 7 ijms-21-07733-f007:**
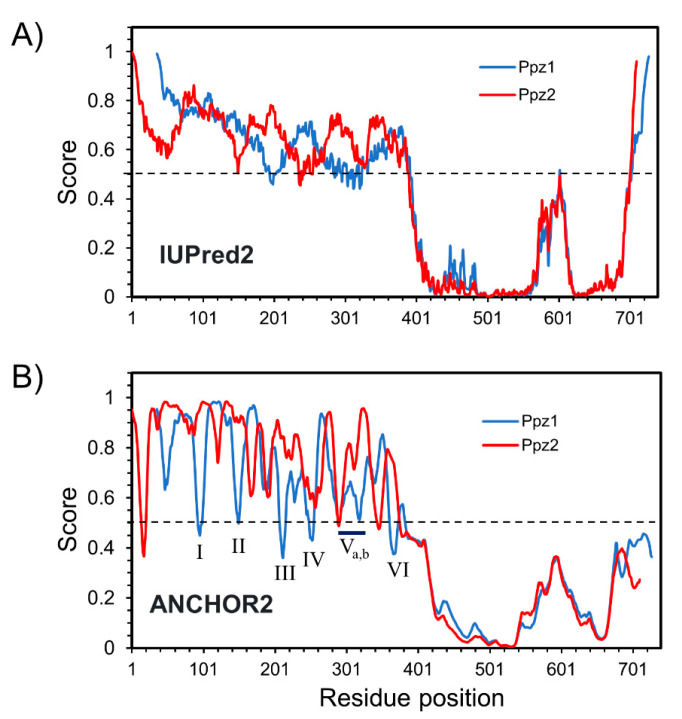
Prediction of intrinsically disordered regions (IDRs) and folding upon binding regions for Ppz1 and Ppz2. (**A**) Prediction of IDRs according the IUPred2 software. Note that the Ppz1 sequence has been shifted 35 residues to allow for the overlapping of the catalytic domain’s profiles. (**B**) Prediction of disordered binding regions according the Anchor2 software. The annotated peaks in Ppz1 (I to VI) are discussed in the text. The 0.5 cut-off (discontinuous line) corresponds to a 5% false positive prediction on IDRs or ordered protein segments.

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
