# Peer review of "The N-Terminal Region of Yeast Protein Phosphatase Ppz1 Is a Determinant for Its Toxicity"

_ijms, 2020, doi:10.3390/ijms21207733_

Round 1

Reviewer 1 Report

The article entitled “The N-terminal region of yeast protein phosphatase 2 Ppz1 is a determinant for toxicity” explores the toxicity of Ppz1 when overexpressed in yeast. This protein and its paralog Ppz2, have a C-terminal catalytic region, related to PP1c phosphatases. Although it was described that phosphatase activity is needed for such toxicity, the authors explore further in the role of the divergent N-terminal extension predicted to be intrinsically disordered.

The work presented clearly demonstrates that such N-terminal extension of Ppz1 plays a key role in its toxicity. Chimeric proteins between Ppz1 and Ppz2 were constructed and expressed, and nicely demonstrated such role in toxicity.

Major point:

The main question that remains to be answer is: Why N-terminal extension is determinant for toxicity? Authors pointed out that Ppz1 toxicity involves the alteration of multiple cellular targets. Interference with protein translation by Ppz1 overexpression plays a key point in yeast toxicity. It is known that an excess of Ppz1 increase Gcn2-dependent increased phosphorylation of eIF2α at Ser-51, down regulating initiation of transcription. Although the effect of Ppz1 on eIF2a must be indirect, using cells that overexpress the chimeric Ppz1:2 protein, phosphorylation of eIF2a at Ser-51 may give important information on the possible role of the N-terminal extension of Ppz1.

Alternatively, expression of the Ppz1:2 protein in Dgcn2 cells may also give some clue. Lack of Gcn2 suppresses the growth defect associate to the overexpression of this chimeric protein?

Since it is known that Ppz1 repress the expression of the Na+/K+-ATPase encoded by the ENA1 gene, another possibility would include quantification of ENA1 (by q-PCR) or the Na/K+-ATPase (by WB) in the strain overexpressing the chimeric Ppz1:2 protein and compare with cells overexpressing Ppz1 and Ppz2.

Minor points:

  • Page 3: “overexpression of Ppz1 results in delayed G1-S transition and accumulation of unbudded cells.” Then, how is it possible that “cells carrying Ppz1 show an increase of about 50% in the budding index value” Shouldn’t be the opposite?

What exactly measure “Budding index”? Fig 2 legend indicated that “Budding index determination. Cells carrying the indicated plasmids (pCM190-derived, HA-tagged versions, ø denotes the empty vector) were processed as described and the percentage of budded and unbudded cells determined.“

  • Page 10: “Analysis of Ppz1 and Ppz2 N-terminal extensions indicates that Ppz1 contains several short regions for which a significant tendency to drive protein-protein interactions is predicted that are not present in Ppz2.” Did the authors attempted to perform CoIP to see differences between Ppz1 and Ppz2-binding proteins? Otherwise, they should expand this area of discussion.

  • Along all the paper, doxycycline was indicated to be used at 100 μg/ml. Is this correct? Usual concentrations are much lower (0.1-5 mg/ml).

Reviewer 2 Report

In this manuscript Calafi et al. further investigate roles of Ppz proteins encoded by two paralog genes in S. cerevisiae, PPZ1 and PPZ2.

PPz proteins play important and interconnected roles on salt tolerance, cell wall integrity and cell cycle progression in budding yeast and these phosphatase are conserved in fungi.

The authors designed experiments to investigate if, as it is the case for Ppz1, overexpression of Ppz2 has toxic effect and analyse contributes of different domains of the two proteins to the observed phenotypes.

They demonstrate that expression of Ppz2 from a multicopy plasmid is not as toxic as expression of Ppz1 under the same conditions. Although their experiments show that in this situations Ppz2 is less abundant than Ppz1, they demonstrate that expression of Ppz2 in these conditions restore WT phenotypes of the relevant mutants.

The mutation of Gly2 in Ppz1 was also studied and interestingly they show that Ppz1G2A doesn’t localize at the cell periphery but retains toxicity if overexpressed, suggesting that targets of toxicity phenotypes are in different cellular compartments.

Moreover, studying different chimeric Ppz1/2 fusions they found that the N-terminal region of Ppz1, in overexpression experiment of these chimeric proteins, is implicated in the observed deleterious effects on cell growth.

Overall the experiments appear to be well performed and the authors’ conclusions are consistent with the presented data.

One concern is about the title, the data support that the N-terminal Ppz1 is a determinant for toxicity when overexpressed as hybrid protein N-terminal-Ppz1-catalytic-domain-Ppz2, and I am not sure that the proposed title includes only this specific situation.

I think it would be nice to add some comments concerning a comparison of the current results with published results of the deletions in the N-terminal regions of Ppz1 (Reference 4: Clotet, J.; Posas, F.; De Nadal, E.; Arino, J. The NH2-terminal extension of protein phosphatase PPZ1 has an essential functional role. J. Biol. Chem. 1996, 271, 26349–26355), in particular with the results of the presented bioinformatics analysis (IUPred2A and Anchor2).

Author Response

In this manuscript Calafi et al. further investigate roles of Ppz proteins encoded by two paralog genes in S. cerevisiae, PPZ1 and PPZ2.

PPz proteins play important and interconnected roles on salt tolerance, cell wall integrity and cell cycle progression in budding yeast and these phosphatase are conserved in fungi.

The authors designed experiments to investigate if, as it is the case for Ppz1, overexpression of Ppz2 has toxic effect and analyse contributes of different domains of the two proteins to the observed phenotypes.

They demonstrate that expression of Ppz2 from a multicopy plasmid is not as toxic as expression of Ppz1 under the same conditions. Although their experiments show that in this situations Ppz2 is less abundant than Ppz1, they demonstrate that expression of Ppz2 in these conditions restore WT phenotypes of the relevant mutants.

The mutation of Gly2 in Ppz1 was also studied and interestingly they show that Ppz1G2A doesn’t localize at the cell periphery but retains toxicity if overexpressed, suggesting that targets of toxicity phenotypes are in different cellular compartments.

Moreover, studying different chimeric Ppz1/2 fusions they found that the N-terminal region of Ppz1, in overexpression experiment of these chimeric proteins, is implicated in the observed deleterious effects on cell growth.

Overall the experiments appear to be well performed and the authors’ conclusions are consistent with the presented data.

Response: We appreciate the positive opinion of the referee and his/her constructive comments.

One concern is about the title, the data support that the N-terminal Ppz1 is a determinant for toxicity when overexpressed as hybrid protein N-terminal-Ppz1-catalytic-domain-Ppz2, and I am not sure that the proposed title includes only this specific situation.

Response: The referee is right in that the title, by itself, could be interpreted in a context wider than the one proposed in our work. Therefore, we have slightly modified it, which now reads: “The N-terminal region of yeast protein phosphatase Ppz1 is a determinant for its toxicity” to avoid this problem.

I think it would be nice to add some comments concerning a comparison of the current results with published results of the deletions in the N-terminal regions of Ppz1 (Reference 4: Clotet, J.; Posas, F.; De Nadal, E.; Arino, J. The NH2-terminal extension of protein phosphatase PPZ1 has an essential functional role. J. Biol. Chem. 1996, 271, 26349–26355), in particular with the results of the presented bioinformatics analysis (IUPred2A and Anchor2).—

Response: We agree with the referee in that it is worthwhile to link both pieces of information. To this end, we have numbered the Anchor2 predicted binding-upon-binding regions (from I to VI) and introduced a paragraph in the Discussion section (page 10) connecting such prediction with the phenotypic information derived from the previously reported deletion analysis of the N-terminal extension of yeast Ppz phosphatases.

Reviewer 3 Report

Overall, the manuscript is well-written and presents important functional aspects of Ppz1 and Ppz2. However, the following scientific information will be necessary to be formally accepted by the journal.

  1. Ppz1 and Ppz2 were tagged by GFP at their C-terminus. Clearly provide the evidence that these C-terminally tagged fusion proteins are functioning properly in place of its endogenous protein.
  2. Ppz1-GFP localization has been shown in the manuscript. provide Ppz2-GFP localization as a comparison. 
  3. Ppz1-Gly2Ala substitution caused drastic disruption to their sub-cellular localization. find one or more residues that play a role in cell toxicity.

Author Response

Overall, the manuscript is well-written and presents important functional aspects of Ppz1 and Ppz2.

Response: We appreciate the positive opinion of the referee and his/her constructive comments.

However, the following scientific information will be necessary to be formally accepted by the journal.

Ppz1 and Ppz2 were tagged by GFP at their C-terminus. Clearly provide the evidence that these C-terminally tagged fusion proteins are functioning properly in place of its endogenous protein.

Ppz1-GFP localization has been shown in the manuscript. provide Ppz2-GFP localization as a comparison.

Response: The referee is right in that we show experimental data (subcellular localization) using a Ppz1-GFP construct. Indeed, as indicated by the referee, we verified in due time that this fusion protein was effective as Ppz1. We are offering now this data as Figure S3 and including this information in the main text. Note, however, that we do not provide any result concerning a Ppz2-GFP construct (we never prepared it), so we feel it is no relevant for the purpose of this paper to investigate the localization of native Ppz2.

Ppz1-Gly2Ala substitution caused drastic disruption to their sub-cellular localization. find one or more residues that play a role in cell toxicity.

Response: Indeed, the identification of specific residues in the N-terminal extension of Ppz1 relevant for cell toxicity is an important issue that we are considering investigating as a next step. However, such investigation will likely take too much time to be completed (several months, at least), so due to the very limited amount of time provided by the journal for revision, we would not be able to offer such results in this paper.

Round 2

Reviewer 1 Report

The authors performed additional experiments that give new information concerning the major point. Minor points have been properly addressed.